# Disinfection of otorhinolaryngological endoscopes with electrolyzed acid water: A cross-sectional and multicenter study

Takayuki Okano[1], Tatsunori Sakamoto[2], Seiji Ishikawa[3], Susumu Sakamoto[4], Masanobu Mizuta[1], Yuji Kitada[1], Keisuke Mizuno[1], Hideki Hayashi[5,6], Youichi Suzuki[7], Takashi Nakano[6,7], Koichi Omori[1]*

1 Department of Otolaryngology, Head and Neck Surgery, Graduate School of Medicine, Kyoto University, Kyoto, Japan, 2 Department of Otolaryngology, Head and Neck Surgery, Faculty of Medicine, Shimane University, Matsue, Shimane, Japan, 3 Ishikawa Clinic, Kyoto, Japan, 4 Department of Otolaryngology, Min-iren Chuo Hospital, Kyoto, Japan, 5 Product Planning Department, Kaigen Pharma Co., Ltd., Osaka, Japan, 6 Project Team for Medical Application of Electrolysis, Central Research Center, Osaka Medical and Pharmaceutical University, Takatsuki, Osaka, Japan, 7 Department of Microbiology and Infection Control, Faculty of Medicine, Osaka Medical and Pharmaceutical University, Takatsuki, Osaka, Japan

* omori@ent.kuhp.kyoto-u.ac.jp

**Data Availability Statement:** All files are available from the KURENAI repository (https://repository.kulib.kyoto-u.ac.jp/dspace/).

## Abstract

Glutaraldehyde, a germicide for reprocessing endoscopes that is important for hygiene in the clinic, might be hazardous to humans. Electrolyzed acid water (EAW) has a broad anti-microbial spectrum and safety profile and might be a glutaraldehyde alternative. We sought to assess EAW disinfection of flexible endoscopes in clinical otorhinolaryngological settings and its *in vitro* inactivation of severe acute respiratory syndrome coronavirus 2 (SARS-CoV-2) and bacteria commonly isolated in otorhinolaryngology. Ninety endoscopes were tested for bacterial contamination before and after endoscope disinfection with EAW. The species and strains of bacteria were studied. The *in vitro* inactivation of bacteria and SARS-CoV-2 by EAW was investigated to determine the efficacy of endoscope disinfection. More than 20 colony-forming units of bacteria at one or more sampling sites were detected in 75/90 microbiological cultures of samples from clinically used endoscopes (83.3%). The most common genus detected was *Staphylococcus* followed by *Cutibacterium* and *Corynebacterium* at all sites including the ears, noses, and throats. In the *in vitro* study, more than $10^7$ CFU/mL of all bacterial species examined were reduced to below the detection limit (<10 CFU/mL) within 30 s after contact with EAW. When SARS-CoV-2 was treated with a 99-fold volume of EAW, the initial viral titer (> $10^5$ PFU) was decreased to less than 5 PFU. Effective inactivation of SARS-CoV-2 was also observed with a 19:1 ratio of EAW to the virus. EAW effectively reprocessed flexible endoscopes contributing to infection control in medical institutions in the era of the coronavirus disease 2019 pandemic.

**Funding:** The authors received no specific funding for this work.

**Competing interests:** I have read the journal's policy and the authors of this manuscript have following competing interests: Protein Clean Sheets and a CLEANTOP KD-1 automatic washer and disinfector used in the present study were provided by Kaigen Pharma, Co., Ltd. and one of the authors, Hideki Hayashi, is an employee of Kaigen Pharma, Co., Ltd. This does not alter our adherence to PLOS ONE policies on sharing date and materials.

## Introduction

Flexible endoscopes, which are used in a wide range of clinical specialties, including otorhinolaryngology and gastroenterology, were initially used for diagnostic purposes only; however, they have recently been increasingly used in clinical practice for screening, diagnosis, and treatments. This has resulted in the increased exposure of patients to endoscopic examinations. Compared to gastrointestinal endoscopy or bronchoscopy, flexible endoscopic examinations in the clinical practice of otorhinolaryngology and head and neck surgery have the following characteristics. First, endoscopic examinations are performed without sedation in a wide range of outpatient departments, including clinics, general hospitals, and tertiary centers. Second, they are often performed without an appointment on the day of the visit, making it difficult to predict the number of endoscopic examinations per day. Finally, the number of endoscopic examinations per day in a single facility is much larger than that of endoscopes owned by that facility. In the field of otorhinolaryngology, endoscopes have direct contact with bodily fluids or mucus containing infectious microorganisms such as otorrhea, nasal discharge, saliva, and sputum, and thus are associated with the risk of contamination. Therefore, knowledge of and techniques related to infection control are essential when using flexible endoscopes in otorhinolaryngology. Indeed, cases of gastrointestinal endoscopy-transmitted accidental infections have been reported since the 1970s in the United States (US) [1, 2], and cases of endoscopy-transmitted infections, including *Helicobacter pylori*, have been reported in Japan [3]. More recently, outbreaks of multidrug-resistant bacterial infections, including duodenal endoscopy-transmitted carbapenem-resistant *Enterobacteriaceae* infection, have been reported in the US [4]. As the demands for hygiene increases, endoscopes should be adequately and efficiently reprocessed to prevent endoscope-transmitted infections. To achieve this, it is necessary to comply with the guidelines and/or manuals on endoscope reprocessing suitable for each specialty, such as gastroenterology [5–8] or otorhinolaryngology [9, 10]. Furthermore, with the increasing demand for disinfection and the prevalence of endoscopy in the field of otorhinolaryngology, a low-cost and safe disinfection method is eagerly awaited for endoscopic reprocessing. This is particularly true because of the emergence of COVID-19 in 2019, the potential risk of transmission through endoscopes used in the nose or throat, and multidrug-resistant bacteria.

Glutaraldehyde, o-phthalaldehyde, and peracetic acid are commonly used high-level disinfectants; however, these chemicals are sometimes hazardous to patients and healthcare professionals [11, 12]. Therefore, ventilation during the endoscopic examination and reprocessing should be considered. A popular alternative to central decontamination is the chlorine dioxides wipe system. Chemical decontamination utilizing wipe systems, such as chlorine dioxide, is acceptable if an endoscopic washer disinfector is unavailable. The chlorine dioxide system is much less expensive but deemed an inferior method of decontamination. The system requires staff to be thoroughly trained and conversant with the technique and introduces the risk of human error [9]. In contrast, electrolyzed acid water (EAW) contains a mixture of oxidizing species. An EAW is generated at the anode of a dual-chamber electrolytic cell with a membrane placed between the anode and cathode by electrolyzing a low-concentration aqueous solution of sodium chloride. At the anode, chloride ions are converted into gaseous chlorine, which then reacts with water to form hypochlorous acid, which plays a major role in disinfection, and results in the formation of EAW at pH 2.7. EAW has a broad antibacterial spectrum and has been shown to remove spore-forming bacteria [13], acid-fast bacteria [14, 15], fungi [16, 17], and blood-borne infection-causing viruses such as hepatitis B virus [18, 19], and human immunodeficiency virus [20]. EAW has also been experimentally proven to be minimally cytotoxic [16]. It is essential to use EAW properly for reprocessing endoscopes because

it is corrosive to metals when the chlorine concentration is high, and it does not provide a sufficient disinfection effect when the chlorine concentration is low. Tsuji et al. demonstrated that EAW eliminates various bacteria and viruses within 5 min in upper gastrointestinal endoscopes, which have a high rate of contamination with bacteria including *Helicobacter pylori*. They also showed the potential of using EAW to solve several clinical problems, such as the toxicity of liquid chemical germicides, prolonged exposure to endoscope disinfection, and the high costs associated with the use of other types of high-level disinfectants [16]. Although EAW has been used to reprocess gastrointestinal endoscopes for more than 20 years and is effective at disinfecting clinically used gastrointestinal endoscopes [16], its efficacy in disinfecting clinically used otorhinolaryngological endoscopes has not been reported.

In the present study, we evaluated the efficacy of EAW for the disinfection of otorhinolaryngological endoscopes in clinical settings. We further conducted a microbiological analysis of the surface of endoscopes, and verified the efficacy of EAW in removing possible pathogenic bacteria related to otorhinolaryngology [21] or severe acute respiratory syndrome coronavirus 2 (SARS-CoV-2) by an *in vitro* study.

## Materials and methods

### Clinical study

The study was conducted as a multicenter research trial with 90 patients who underwent flexible endoscopic examinations in the Department of Otorhinolaryngology at three medical sites: a primary care clinic, a secondary care hospital, and a tertiary care center (30 patients per site).

### Microbiological tests

First, to verify contamination, samples were collected before the manual cleaning of used otorhinolaryngological endoscopes. For non-channeled endoscopes, samples were collected at two sites: the overall area of the flexible tube and bending tip inserted into the patient (insertion site) (A-1) and the overall area of the control handle and angulation knob operated by the examiner (operation site) (A-2). For channeled endoscopes with a suction/forceps channel, samples were collected at three sites, including A-1, A-2, and the inside of the operating channel (A-3) (Fig 1). A sample was collected at A-1 and A-2 by swabbing the sampling site with a wet bacteriological swab (ST25-100, ELMEX Co., Ltd., Tokyo, Japan) ten times, and the swab was stored in a sterile tube. At A-3, the suction/forceps channel was flushed with 20 mL of sterile normal saline from the forceps channel inlet, and the saline was collected in a sterile tube at the forceps channel outlet and stored. Subsequently, the endoscopes were cleaned in accordance with the recommendation on reprocessing techniques of flexible otorhinolaryngological endoscopes [10] as follows:

1. After sampling, the outside surface of the endoscope was wiped with Tanpaclean wipes (Kaigen Pharma Co., Ltd., Osaka, Japan) and gauze was soaked in a cleaning solution.

2. The outside surface of the endoscope was cleaned with a sponge and detergent containing protease enzymes, followed by rinsing with water to remove residual blood, body fluids, and proteins.

3. For channeled endoscopes with a suction/forceps channels, the inside of the channel was cleaned with a brush and flushed with at least 200 mL of cleaning solution to remove contaminants.

4. The enzyme detergent was thoroughly removed by washing with tap water, and the inside of the channel was flushed with water to wash out the detergent.

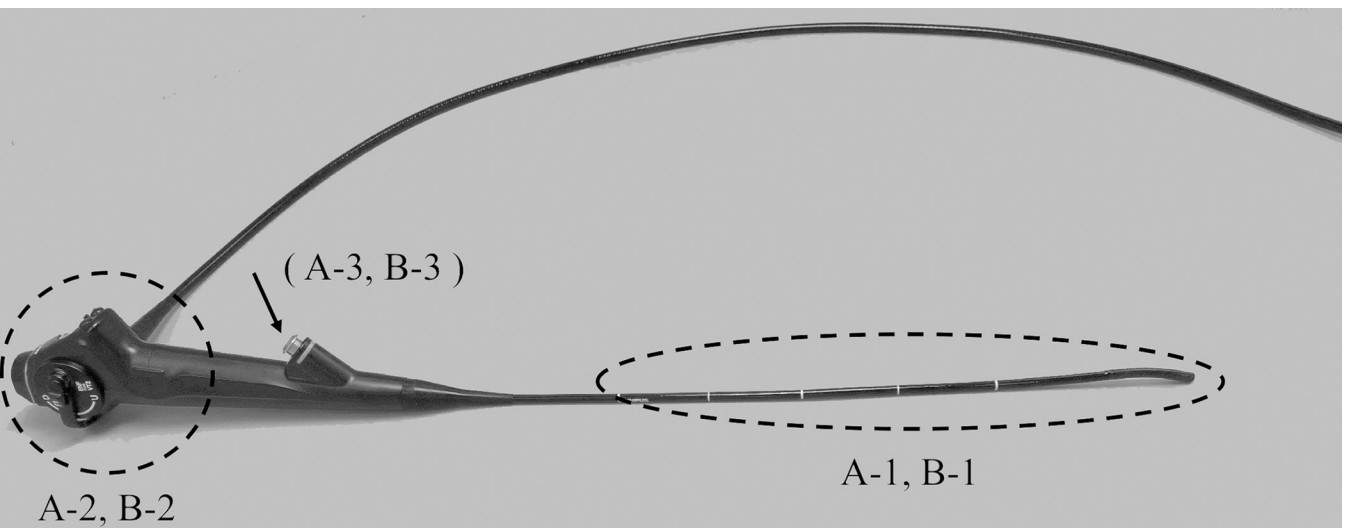

**Fig 1. Sites of microbiological sampling in otorhinolaryngological endoscopes.** Microbiological samples were collected at two or three sites for non-channeled endoscopes or channeled endoscopes with a suction/forceps channel, respectively, before (A-1, 2, 3) and after (B-1, 2, 3) cleaning and disinfection with EAW using a Cleantop KD-1. A-1 and B-1: overall area of the flexible tube and bending tip inserted into the patient (insertion site); A-2 and B-2: overall area of the control handle and angulation knob operated by the examiner (operation site); A-3 and B-3: inside of the operating channel.

After manual cleaning, the endoscopes were disinfected using a Cleantop KD-1 (Kaigen Pharma Co., Ltd.), a disinfector designed to disinfect flexible endoscopes with EAW. The effective chlorine concentration of Cleantop KD-1 was between 10 and 40 ppm, and a course of disinfection took 90 s. During the cleaning and disinfection of the endoscopes, personal protective equipment such as masks, gloves, goggles, and gowns were worn appropriately, and adequate ventilation was provided in the room.

Immediately after cleaning and disinfection, contaminated samples were collected from the endoscopes in the same manner as before cleaning and disinfection. For non-channeled endoscopes, samples were collected at two sites: the overall area of the flexible tube and bending tip inserted into the patient (insertion site) (B-1), and the overall area of the control handle and angulation knob operated by the examiner (operation site) (B-2). For channeled endoscopes, samples were collected at three sites: B-1, B-2, and inside the operating channel (B-3) (Fig 1).

All samples collected were promptly cultured for aerobic bacteria, anaerobic bacteria, and fungi at the Japan Microbiological Clinic Co., Ltd. (Kanagawa, Japan), followed by bacterial culture to calculate the proportion of effectively disinfected endoscopes and identify species and strains of bacteria present. The analysis included the endoscopes used in patients with more than 20 colony-forming units (CFU) of bacteria at one or more sampling sites before disinfection. Effective disinfection was defined as a bacterial count of ≤20 CFU per sampling site and no detection of indicator microorganisms after disinfection based on the European Society of Gastrointestinal Endoscopy and European Society of Gastroenterology Nurses and Associates Guidelines for Quality Assurance in Reprocessing: Microbiological Surveillance Testing in Endoscopy [22].

### *In vitro* studies for the antibacterial and antiviral response of EAW

A dual-chamber electrolytic cell was equipped with platinum-coated titanium electrodes (11 × 19 cm each, CT-501S, Tanaka Kikinzoku Kogyo K. K., Tokyo, Japan) with the chambers separated by a cationic membrane (Nafion 424, DuPont, DE, USA) used to produce EAW. EAW was obtained at the anode side by electrolyzing 500 mL of 0.1% sodium chloride solution

for 6–7 min at a constant voltage of 24 V. The free (available) chlorine concentration was measured using a chlorine meter (DP-3F, Kasahara Chemical Instruments Corp., Saitama, Japan), and the pH was measured using a pH meter (HM-31P, DKK-TOA Corp., Tokyo, Japan). The effective chlorine concentration, pH, and oxidation-reduction potential of EAW used in bacteria experiments were adjusted to 20.66 ± 0.18 ppm, 2.60 ± 0.00, and 1133.7 ± 3 mV, respectively. For SARS-CoV-2 experiments, they were adjusted to 10.57 ± 0.34 ppm, 2.61 ± 0.01, and 1119.5 ± 6.2 mV, respectively.

## Bacterial culture and bactericidal effects of EAW

The bacteria used in the tests were *Staphylococcus aureus* NBRC13276, *Streptococcus pneumoniae* NBRC102642, *Streptococcus pyogenes* GTC262, *Klebsiella pneumoniae* NBRC3512, *Haemophilus influenzae* IID983, and clinical isolates of β-lactamase-negative ampicillin-resistant *Haemophilus influenzae* (BLNAR) nos.1 and no.2. *S. pyogenes* GTC262 was provided by the Center for Conservation of Microbial Genetic Resources, Gifu University, through the National Bioresource Project (NBRP) of the Ministry of Education, Culture, Sports, Science and Technology (MEXT), Japan. *H. influenzae* IID983 was provided by the Institute of Medical Science, The University of Tokyo, through the NBRP of MEXT, Japan. *S. pyogenes* was cultured on blood agar (Nissui Pharmaceutical Co., Ltd., Tokyo, Japan) in an anaerobic environment. *S. pneumoniae* was cultured on blood agar (Nissui Pharmaceutical Co., Ltd.), *S. aureus* and *K. pneumoniae* were cultured on heart infusion agar (Eiken Chemical Co., Ltd., Tokyo, Japan), and *H. influenzae* was cultured on chocolate agar (Nissui Pharmaceutical Co., Ltd.) in an aerobic environment. Bacterial cells were cultured on plates for 16 h and collected. To prepare bacterial suspensions for testing, the bacterial cells were collected and suspended in distilled water to adjust a concentration same as the turbidity of McFarland Standard No. 2. Subsequently, EAW and the bacterial suspension were mixed in a 1:1 ratio. After they were in contact for a designated period, free chlorine was neutralized with an equal volume of 1% bovine serum albumin solution [23]. To quantify residual viable bacterial cells, a specimen neutralized for a designated period was serially diluted and spread on plates. The plates were cultured overnight at 37°C, and the number of colonies was determined to calculate the number of residual viable bacterial cells.

## EAW treatment of SARS-CoV-2

Two isolates of SARS-CoV-2 (WK-521 [provided by the National Institute of Infectious Diseases, Tokyo, Japan] and OMC-510) were amplified and titrated using Vero E6/TMPRSS2 cells as described previously [24]. Infectious titers of virus stocks of WK-521 and OMC-510 were $1.2 \times 10^7$ plaque-forming units (PFU)/mL and $5.0 \times 10^7$ PFU/mL. EAW was added to each virus stock at a 990 μL:10 μL or 950 μL:50 μL ratio, and reacted at room temperature for 1 min. Subsequently, 100 μL of Dulbecco's modified Eagle's medium supplemented with 10% fetal bovine serum was added to the mixture to neutralize the EAW and further incubated at room temperature for 5 min. After incubation, the viral titer in the treated samples were determined as PFU/mL. Sodium chloride solution (0.1%) was used as a control.

## Ethical approval

This study was performed in accordance with the ethical standards of the Declaration of Helsinki of 1975 and its later amendments or comparable ethical standards. The study protocol of the present study was approved by the Institutional Review Boards and Ethics Committees of Kyoto University Hospital and Kyoto University Graduate School of Medicine (R2386). Verbal informed consent for participation was obtained from patients visiting the Department of

Otorhinolaryngology outpatient facility using an opt-out methodology because this study does not harm the patients in any way, including direct microbiological or blood sampling.

## Results

### General characteristics of patients and endoscopes

The patients included in the present study had a mean age of 59 years with a range from 1 to 92 years, and consisted of 50 males and 40 females. Of those with primary diseases that required endoscopic examination, 55 patients had laryngopharyngeal disease (pharyngitis in 9, laryngeal cancer in 4, dysphagia in 3, tongue cancer in 3, hypopharyngeal cancer in 3, oropharyngeal cancer in 3, nasopharyngeal cancer in 2, and 28 with other), 29 patients had nasal disease (sinusitis in 20, epistaxis in 5, and other condition in 4), and 6 patients had ear disease (otitis media in 4, otitis externa in 1, and ear fullness caused by sinusitis in 1) (Table 1). Regarding the non-channeled endoscopes used, Pentax VNL-90s was used in 30 patients, Olympus ENF-VH in 25, Olympus ENF-V3 in 16, Olympus ENF TYPE VQ in 8, and Olympus ENF TYPE V2 in 1. Channeled endoscopes used in this study were the Olympus ENF TYPE VT2 in 6 patients and Olympus ENF-VT3 in 4 patients.

### Endoscope disinfection using EAW in clinical settings

After the microbiological culture of clinically used endoscopes, more than 20 CFU of bacteria at one or more sampling sites was detected in 75 of 90 patients (83.3%) (Table 2). More specifically, microorganisms were detected directly after clinical use without disinfection in 13 of 90 patients (14.4%), 74 of 90 patients (82.2%), and 9 of 10 patients (90.0%) at the endoscopic operation site, endoscopic insertion site, and inside the endoscopic suction/forceps channel, respectively. After cleaning and disinfection using EAW, microorganisms were detected in only one of 75 endoscopes inside the endoscopic suction/forceps channel with a value of 30 CFU, which represents a $> \times 10^5$ reduction in a clinical environment. After reprocessing with the EAW, the remaining 74 samples showed no further bacterial contamination.

**Table 1. Primary diseases requiring endoscopic examination.**

| Examination sites | Identified bacterial species | Primary diseases | | | Total |
|---|---|---|---|---|---|
| | | Tertiary care center | Secondary care hospital | Primary care clinic | |
| Throat in 55 patients | pharyngitis | 1 | - | 8 | 9 |
| | laryngeal cancer | 2 | 2 | - | 4 |
| | dysphagia | 1 | 2 | - | 3 |
| | tongue cancer | 3 | - | - | 3 |
| | hypopharyngeal cancer | 3 | - | - | 3 |
| | oropharyngeal cancer | 3 | - | - | 3 |
| | nasopharyngeal cancer | 2 | - | - | 2 |
| | Other | 2 | 1 | - | 3 |
| Nose in 29 patients | sinusitis | - | 6 | 14 | 20 |
| | epistaxis | - | 5 | - | 5 |
| | Other | 1 | 2 | 1 | 4 |
| Ear in 6 patients | otitis media | - | 2 | 2 | 4 |
| | otitis externa | - | - | 1 | 1 |
| | ear fullness caused by sinusitis | - | 1 | - | 1 |

**Table 2. Characteristics of patients and endoscopes included in this study.**

| Study site name | Number of patients | Number of endoscopes used in patients and included in analysis [a] | Number of endoscopes used in patients with effective disinfection [b] | Effective disinfection rate (%) [c] | Sampling site | Number of samples | Detection of microorganisms [d] | |
|---|---|---|---|---|---|---|---|---|
| | | | | | | | Before disinfection | After disinfection |
| Tertiary care center | 30 | 28 | 28 | 100 | Operation site | 30 | 10 | 0 |
| | | | | | Insertion site | 30 | 28 | 0 |
| | | | | | Inside the suction/forceps channel | 10 | 10 | 0 |
| Secondary care hospital | 30 | 25 | 24 | 96 | Operation site | 30 | 6 | 0 |
| | | | | | Insertion site | 30 | 25 | 0 |
| | | | | | Inside the suction/forceps channel | 1 | 1 | 1 |
| Primary care clinic | 30 | 22 | 22 | 100 | Operation site | 30 | 5 | 0 |
| | | | | | Insertion site | 30 | 23 | 0 |
| | | | | | Inside the suction/forceps channel | 0 | 0 | 0 |
| Total | 90 | 75 | 74 | 98.7 | | | | |

[a] Endoscopes used in patients with more than 20 CFU of bacteria at one or more sampling sites before disinfection.

[b] Endoscopes with a bacterial count ≤20 CFU per sampling site and no detection of indicator microorganisms after disinfection.

[c] Effective disinfection rate (%) = 100 × (number of endoscopes with effective disinfection/number of endoscopes included in the analysis).

[d] Endoscopes with more than 1 CFU of bacteria at a sampling site.

CFU: colony-forming units.

## *Staphylococcus* species are the most common potential pathogen in otorhinolaryngological endoscopes

The most common genus of strains detected was *Staphylococcus*, followed by *Cutibacterium* and *Corynebacterium* at all sites including the ear, nose, and throat (Table 3). By test location, *Staphylococcus* and *Cutibacterium* were most commonly detected at all test locations, with no differences among primary, secondary, and tertiary medical institutes. In endoscopes with ineffective disinfection, *S. aureus* was detected before and after cleaning and disinfection. No fungi were detected in this study.

## EAW has a wide range of *in vitro* activity as a germicide against various microorganisms

In the *in vitro* study, EAW was added to each bacterial suspension in McFarland No. 2 standard, and viable counts were determined at each specific time point to determine the bactericidal activity of EAW (Table 4). To determine the relationship between drug resistance and disinfection resistance, a similar test was performed using clinical isolates of BLNAR. EAW reduced the viable counts of all bacteria to $<10^7$ within 30 s and successfully removed drug-resistant bacteria. No controls were used in this study, however, we confirmed that microorganisms not treated with EAW survived at more than $10^7$ under the same culture conditions.

## EAW has inactivating activity against SARS-CoV-2

After contacting the SARS-CoV-2 suspensions for 1 min, EAW inactivated SARS-CoV-2 by ≥99.9% in all virus suspensions at different mixture ratios (Table 5). Furthermore, a mixture

**Table 3. Bacterial species and location of isolation.**

| Examination sites | Identified bacterial species | Number of strains detected [a] | | | Total |
|---|---|---|---|---|---|
| | | Tertiary care center | Secondary care hospital | Primary care clinic | |
| Throat in 55 patients | *Staphylococcus* | 40 | 15 | 12 | 67 |
| | *Cutibacterium* | 21 | 9 | 3 | 33 |
| | *Corynebacterium* | 6 | 2 | 2 | 10 |
| | *Streptococcus* | 6 | 1 | - | 7 |
| | *Pseudomonas* | 4 | - | - | 4 |
| | *Bacillus* | 2 | 1 | - | 3 |
| | *Rothia* | 2 | 1 | - | 3 |
| | *Serratia* | 2 | 1 | - | 3 |
| | *Klebsiella* | 1 | - | 1 | 2 |
| | *Other* | 6 | 2 | 1 | 9 |
| Nose in 29 patients | *Staphylococcus* | 2 | 9 | 12 | 23 |
| | *Cutibacterium* | - | 4 | 4 | 8 |
| | *Corynebacterium* | 1 | 5 | 2 | 8 |
| | *Bacillus* | 1 | - | 1 | 2 |
| | *Citrobacter* | - | 1 | - | 1 |
| | *Klebsiella* | - | 1 | - | 1 |
| | *Proteus* | - | | 1 | 1 |
| Ear in 6 patients | *Staphylococcus* | - | 3 | 2 | 5 |
| | *Cutibacterium* | - | 2 | 1 | 3 |
| | *Bacillus* | - | - | 1 | 1 |

[a] Multiple strain were detected in some samples.

ratio of 99:1 was associated with a greater reduction in viral load than a ratio of 19:1 and ≥99.9% inactivation.

## Discussion

In this study, microorganisms were detected once in a channeled endoscope used in 90 patients after disinfection of the inside of the channel. Several factors may have limited the efficacy of the disinfection procedures. Potential causes included (1) contamination by airborne

**Table 4. Reduction of bacteria by EAW.**

| Bacterial strain | Initial density (CFU/mL) | After EAW contact (30 s) |
|---|---|---|
| *Streptococcus pneumoniae* NBRC102642 | $1.1 \times 10^7$ | n.d. |
| *Streptococcus pyogenes* GTC262 | $7.9 \times 10^7$ | n.d. |
| *Staphylococcus aureus* NBRC13276 | $3.8 \times 10^7$ | n.d. |
| *Klebsiella pneumoniae* NBRC3512 | $9.8 \times 10^7$ | n.d. |
| *Haemophilus influenzae* IID983 | $2.9 \times 10^7$ | n.d. |
| Clinical isolates of β-lactamase-negative ampicillin-resistant *Haemophilus influenzae* (BLNAR) no.1 | $3.9 \times 10^7$ | n.d. |
| Clinical isolates of β-lactamase-negative ampicillin-resistant Haemophilus influenzae (BLNAR) no. 2 | $1.0 \times 10^8$ | n.d. |

n.d.: below the limit of detection (<10 CFU/mL); EAW: electrolyzed acid water; CFU: colony-forming units; BLNAR: β-lactamase-negative ampicillin-resistant *Haemophilus influenzae*.

**Table 5. Inactivating activity of EAW against SARS-CoV-2.**

| Isolates | Titer of virus stock [a] | Mixing ratio virus: EAW | Titer of EAW-contacted virus [a] | Titer of control-contacted virus [a] |
|---|---|---|---|---|
| SARS-CoV-2/WK-521 | $1.2 \times 10^7$ | 1:19 | 50 | $7.0 \times 10^5$ |
| | | 1:99 | n.d.[†] | $1.3 \times 10^5$ |
| SARS-CoV-2/OMC-510 | $5.0 \times 10^7$ | 1:19 | $2.2 \times 10^3$ | $3.4 \times 10^6$ |
| | | 1:99 | n.d. | $5.0 \times 10^5$ |

[a] PFU/mL. n.d.: below the limit of detection (< 5 PFU/mL); EAW: electrolyzed acid water; PFU: plaque-forming units; SARS-CoV-2: severe acute respiratory syndrome coronavirus 2.

bacteria in the environment or normal skin flora of a sampler during sampling, (2) inadequate cleaning and disinfection of the endoscope in a washer disinfector, and (3) the effect of biofilms. Contamination, as described in (1), cannot be completely ruled out for the test system in this study. Inadequate cleaning and disinfection as described in (2) should be ruled out because EAW is sufficiently effective enough to eliminate *S. aureus*, the species detected after cleaning and disinfection. The effect of biofilms, as described in (3), cannot be ruled out as a possible cause of residual bacteria after disinfection with EAW. These findings indicate the importance of properly storing endoscopes after cleaning and disinfection. The channeled endoscope in which residual bacteria were detected had been stored in a carrying case for more than a month after its last use rather than in a storage cabinet. Because channeled endoscopes with a suction/forceps channels might be used infrequently in community hospitals and clinics in Japan, they might remain unused for more than a month after their last use, which was the case in the present study. Carrying cases for endoscopes are not sufficiently ventilated and provide conditions favorable for pathogen growth. Particular caution should be exercised when storing a channeled endoscope, because water tends to remain in the channel. The guidance states that no endoscope should be stored in a coiled position nor in a carrying case [10].

The present clinical study demonstrated that EAW inactivated the gram-positive bacteria *S. aureus*, *S. pyogenes*, and *S. pneumoniae* and the gram-negative bacteria *H. influenzae* and *K. pneumoniae*, which cause otorhinolaryngological diseases [21], although these strains were not often detected. Our *in vitro* data also showed that EAW exhibited a wide range of microbicidal activities against bacteria frequently isolated from otorhinolaryngological sites. No fungi were detected in the present study; however, Urata et al. demonstrated that EAW has microbicidal activity against *Candida albicans*, one of the major fungal species isolated from the oral cavity or throat [17]. These findings suggest the suitability of EAW for disinfection of endoscopes in a otorhinolaryngological environment.

In our *in vitro* study on the inactivation of SARS-CoV-2, EAW inactivated SARS-CoV-2 by ≥99.9% within 1 min at an effective chlorine concentration of 10 ppm, and a mixture ratio of 99:1 was associated with greater inactivation than a ratio of 19:1. This suggests that the protein content in the virus suspensions affected the inactivation activity of EAW. Because more than 5 L of EAW is used to disinfect a flexible endoscope in clinical settings, a mixture ratio of 99:1 may more closely mimic actual use in clinical settings. In addition, the thorough removal of contaminants, such as proteins, by conventional preliminary manual cleaning is important to ensure the consistent disinfection performance of EAW in clinical settings. EAW has been used to disinfect SARS-CoV-2 as an alternative to alcohol. Takeda et al. reported that the viricidal activity of EAW against SARS-CoV-2 depended on the amount of free available chlorine, indicating that an acidic solution without free available chlorine does not inactivate SARS-CoV-2 over a short period [25]. These findings were supported by an *in vitro* study by Xiling

et al., which showed that free available chlorine at 1,000 mg/L inactivated SARS-CoV-2 in 30 s [26].

Regarding the limitations of this clinical study, we designed it to detect aerobic bacteria, anaerobic bacteria, and fungi, however, it is possible that not all microorganisms could be detected. Considering this, it is sufficient to confirm the trend of microorganisms detected in the field of otorhinolaryngology and the disinfection effect of EAW. Taken together, EAW is a safe, broad-spectrum disinfectant that can be used for reprocessing endoscopes during the coronavirus disease 2019 pandemic.

## Conclusion

This study demonstrated that the EAW system for reprocessing flexible endoscopes appears to be an ideal disinfection system with a broad anti-microbial spectrum with both bactericidal and viricidal effects and safety profiles. EAW does not harm human tissues, therefore, it could be an alternative to commonly used high-level disinfectants including glutaraldehyde, o-phthalaldehyde, and peracetic acid. Because EAW disinfectant production requires only salt, tap water, and electricity, and EAW loses its oxidative and acidic properties when exposed to the environment, the use of EAW in the disinfection of otorhinolaryngological endoscopes should contribute to infection control in medical institutions from the viewpoints of safety for medical staffs and environmental friendliness in order to overcome supply crisis during the coronavirus disease 2019 pandemic, indicating sustainable systems and enabling safe continuity.

## Acknowledgments

We thank Ms. Yuriko Shibata at the Central Clinical Laboratory of Osaka Medical and Pharmaceutical University Hospital for kindly providing the clinical isolates of the BLNAR strains of *H. influenzae*.

## Author Contributions

**Conceptualization:** Tatsunori Sakamoto, Hideki Hayashi, Takashi Nakano, Koichi Omori.

**Data curation:** Takayuki Okano, Seiji Ishikawa, Susumu Sakamoto, Masanobu Mizuta, Yuji Kitada, Keisuke Mizuno, Youichi Suzuki.

**Formal analysis:** Hideki Hayashi, Youichi Suzuki.

**Resources:** Hideki Hayashi.

**Supervision:** Takashi Nakano, Koichi Omori.

**Validation:** Takashi Nakano, Koichi Omori.

**Writing – original draft:** Takayuki Okano, Hideki Hayashi.

**Writing – review & editing:** Takayuki Okano.

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
