## [Decision Letter · Decision Letter 0]

16 Aug 2022

PONE-D-22-18329Disinfection of otorhinolaryngological endoscopes with electrolysed acid water: a cross-sectional and multicentre studyPLOS ONE

Dear Dr. Omori,

Thank you for submitting your manuscript to PLOS ONE. After careful consideration, we feel that it has merit but does not fully meet PLOS ONE’s publication criteria as it currently stands. Therefore, we invite you to submit a revised version of the manuscript that addresses the points raised during the review process.

We look forward to receiving your revised manuscript.

Kind regards,

Awatif Abid Al-Judaibi, PhD

Academic Editor

PLOS ONE

Journal Requirements:

"I have read the journal's policy and the authors of this manuscript have the following competing interests: Protein Clean Sheets and a CLEANTOP KD-1 automatic washer and disinfector used in the present study were provided by Kaigen Pharma, Co., Ltd. and one of the authors, Hideki Hayashi, is an employee of Kaigen Pharma, Co., Ltd."

Additional Editor Comments:

Reviewer 1:

1- Significant differences between A-1, A-2, and A-3 . sampling sites should be mentioned.

2- There are some typos and grammatical errors which should be revised.

3- This sentence is not correct "The verbal

210 informed consent for participation was obtained form patients visiting the outpatient facility of the

211 Department of Otorhinolaryngology using an opt-out methodology because this study does not

212 harm the patients in any way including direct microbiological sampling or blood sampling"

4- I couldn't find any of the references from the 21, 22 and 23

Reference correction is necessary

Reviewer 2:

Authors have reported effective use of Electrolyte acid water (EAW) in disinfecting otorhinolaryngological endoscopes, which is of clinical significance. Authors have presented sufficient data in support of research question keeping limitation of study in account. Manuscript can be accepted for publication after making amendments as suggested.

Some specific comments/reviews to incorporate/amend are as follows:

1- Abstract: Authors have summarized key findings appropriately. However, in vitro inactivation of bacteria and SARS-CoV-2 need to be presented consistently in number/percentage or CFU to enhance reader's understanding.

2-Introduction: Significance of using EAW for disinfection of otorhinolaryngological endoscopes for bacteria has been presented appropriately. however, authors need to include significance of using EAW for SARS-CoV-2 as well briefly.

2- Methods: Authors used appropriate methodology to address the research question. However, following amendments 13are suggested:

Page 8, lines 138-139: Authors either need to include time/concentration for disinfection using Cleantop-KD-1 or reference

for the procedure.

Page 11, lines 190-191: "Overnight" incubation time need to be rephrased in terms of "Hours" for clarity.

3- Results: Authors have presented results appropriately using tables facilitating reader's understanding. Suggested amendment:

Page 13, lines 220-221: "29 patients had nasal disease....." needs correction as breakdown adds up to 30.

Page 14, lines 234- 235: " ...only in 1 of 75 patients...." needs to be corrected as " ...only in 1 of 75 endoscopes...."

4-Discussion: Overall needs to be rephrased. Few amendments suggested are:

Page 18-19, line 286- 301, "Because otorhinolaryngology ........high-level disinfection." is a repeat from introduction

need to be deleted.

Page 19-20, para 2, line 309-320: Needs rephrasing for clarity. Also, reference of table needs to be removed from

discussion.

5- Conclusion: It is vague and needs rephrasing keeping context of study in view to highlight the significance of study.

Reviewer 3:

1. Lines 90 - 93: Despite there is no study that presents the disinfection efficacy of EAW in clinically used otorhinolaryngological endoscopes, authors have mentioned about the effectiveness of using EAW in disinfecting clinically for gastrointestinal endoscopes. Authors should highlight their significant findings and contributions.

2. Introduction can be improved by adding a few sentences about the hypochlorous acid since it is the primary antibacterial agent in the study.

3. Line 95: "Fig 1. Schematic of generation of electrolysed acid water" is not necessary to be included in the manuscript, as it is well known.

4. Lines 116 –148: it is helpful to provide an image (as supporting Information file) of the endoscopy instruments with two sites (A-1 to A-3, B-1 to B-3) to aid the understanding of this part.

5. Line 149 and line 340: this a study designed to culture aerobic and anaerobic bacteria as well as fungi but there is limited discussion about fungi being observed or detected or fungicidal effect, please explain more.

6. Line 171: the subheading "bacteria and culture" is more generic while the paragraph discusses EAW treatment; please edit the subheading to reflect all the aspects of the method on bacterial culture and bactericidal effects of EAW.

7. Lines 216- 226: A table is preferred here given the amount of data being presented.

8. Line 270: What is the control sample for the data presented in the Table 3?

9. Full name of bacterial strains should be mentioned in the (Table 3) as mentioned in from line 172-175 have to be.

10. Lines 258 – 301: Discussion can be improved to avoid repetition with introduction section.

11. The conclusion is supported by the results for EAW treatment of SARS-CoV-2 but not for bacteria. Therefore, this section must be improved and highlight why EAW can be an ideal disinfectant (bactericidal and virucidal effects) and use it at present as regarded to be safe and environmentally friendly to overcome supply crisis during pandemic emergency and enable the safe continuity.

12. Line 183: influenzae, make the last letter (e) italic as well.

13. Line 210: form or from

Reviewers' comments:

Reviewer's Responses to Questions

**Comments to the Author**

1. Is the manuscript technically sound, and do the data support the conclusions?

Reviewer #1: Yes

Reviewer #2: Partly

Reviewer #3: Partly

2. Has the statistical analysis been performed appropriately and rigorously? 

Reviewer #1: I Don't Know

Reviewer #2: N/A

Reviewer #3: N/A

3. Have the authors made all data underlying the findings in their manuscript fully available?

Reviewer #1: Yes

Reviewer #2: Yes

Reviewer #3: Yes

4. Is the manuscript presented in an intelligible fashion and written in standard English?

Reviewer #1: Yes

Reviewer #2: No

Reviewer #3: Yes

5. Review Comments to the Author

Reviewer #1: Please see the following published research:

1- https://www.researchgate.net/profile/Raad-Hasan/publication/357303354_Detection_of_fimH_kpsMTII_hlyA_and_traT_genes_in_Escherichia_coli_isolated_from_Iraqi_patients_with_cystitis/links/61c5a20bb8305f7c4bf970a1/Detection-of-fimH-kpsMTII-hlyA-and-traT-genes-in-Escherichia-coli-isolated-from-Iraqi-patients-with-cystitis.pdf

2- https://www.researchgate.net/deref/https%3A%2F%2Fpubmed.ncbi.nlm.nih.gov%2F35096341%2F

3- https://www.researchgate.net/deref/https%3A%2F%2Fwww.cabdirect.org%2Fglobalhealth%2Fabstract%2F20193498480

4- https://www.researchgate.net/deref/https%3A%2F%2Fworldresearchersassociations.com%2FBiotechSpecialIssueMarch2019%2F7.pdf

Reviewer #2: Authors have reported effective use of Electrolyte acid water (EAW) in disinfecting otorhinolaryngological endoscopes, which is of clinical significance. Authors have presented sufficient data in support of research question keeping limitation of study in account. Manuscript can be accepted for publication after making amendments as suggested.

Some specific comments/reviews to incorporate/amend are as follows:

1- Abstract: Authors have summarized key findings appropriately. However, in vitro inactivation of bacteria and SARS-CoV-2 need to be presented consistently in number/percentage or CFU to enhance reader's understanding.

2-Introduction: Significance of using EAW for disinfection of otorhinolaryngological endoscopes for bacteria has been presented appropriately. however, authors need to include significance of using EAW for SARS-CoV-2 as well briefly.

2- Methods: Authors used appropriate methodology to address the research question. However, following amendments 13are suggested:

Page 8, lines 138-139: Authors either need to include time/concentration for disinfection using Cleantop-KD-1 or reference

for the procedure.

Page 11, lines 190-191: "Overnight" incubation time need to be rephrased in terms of "Hours" for clarity.

3- Results: Authors have presented results appropriately using tables facilitating reader's understanding. Suggested amendment:

Page 13, lines 220-221: "29 patients had nasal disease....." needs correction as breakdown adds up to 30.

Page 14, lines 234- 235: " ...only in 1 of 75 patients...." needs to be corrected as " ...only in 1 of 75 endoscopes...."

4-Discussion: Overall needs to be rephrased. Few amendments suggested are:

Page 18-19, line 286- 301, "Because otorhinolaryngology ........high-level disinfection." is a repeat from introduction

need to be deleted.

Page 19-20, para 2, line 309-320: Needs rephrasing for clarity. Also, reference of table needs to be removed from

discussion.

5- Conclusion: It is vague and needs rephrasing keeping context of study in view to highlight the significance of study.

Reviewer #3: Manuscript Number: PONE-D-22-18329

Type: Research Article Title: Disinfection of otorhinolaryngological endoscopes with electrolysed acid water: a cross-sectional and multicentre study

Authors: Takayuki Okano1, Tatsunori Sakamoto2, Seiji Ishikawa3, Susumu Sakamoto4, Masanobu Mizuta1, Yuji Kitada1, Keisuke Mizuno1, Hideki Hayashi5,6, Youichi Suzuki7, Takashi Nakano6,7, Koichi Omori1*

Summary: The article describes the practical steps involved in preparing and evaluating electrolyzed acidic water for the disinfection of otorhinolaryngological endoscopes in clinical settings, and an in vitro study of the efficacy of EAW at removing possible pathogenic bacteria related to otorhinolaryngology and severe acute respiratory syndrome coronavirus 2 (SARS-CoV-2).

Overall, the research design is appropriate, the methods adequately described, the results clearly presented and all cited references relevant to the research.

Comments to authors:

1. Lines 90 - 93: Despite there is no study that presents the disinfection efficacy of EAW in clinically used otorhinolaryngological endoscopes, authors have mentioned about the effectiveness of using EAW in disinfecting clinically for gastrointestinal endoscopes. Authors should highlight their significant findings and contributions.

2. Introduction can be improved by adding a few sentences about the hypochlorous acid since it is the primary antibacterial agent in the study.

3. Line 95: "Fig 1. Schematic of generation of electrolysed acid water" is not necessary to be included in the manuscript, as it is well known.

4. Lines 116 –148: it is helpful to provide an image (as supporting Information file) of the endoscopy instruments with two sites (A-1 to A-3, B-1 to B-3) to aid the understanding of this part.

5. Line 149 and line 340: this a study designed to culture aerobic and anaerobic bacteria as well as fungi but there is limited discussion about fungi being observed or detected or fungicidal effect, please explain more.

6. Line 171: the subheading "bacteria and culture" is more generic while the paragraph discusses EAW treatment; please edit the subheading to reflect all the aspects of the method on bacterial culture and bactericidal effects of EAW.

7. Lines 216- 226: A table is preferred here given the amount of data being presented.

8. Line 270: What is the control sample for the data presented in the Table 3?

9. Full name of bacterial strains should be mentioned in the (Table 3) as mentioned in from line 172-175 have to be.

10. Lines 258 – 301: Discussion can be improved to avoid repetition with introduction section.

11. The conclusion is supported by the results for EAW treatment of SARS-CoV-2 but not for bacteria. Therefore, this section must be improved and highlight why EAW can be an ideal disinfectant (bactericidal and virucidal effects) and use it at present as regarded to be safe and environmentally friendly to overcome supply crisis during pandemic emergency and enable the safe continuity.

12. Line 183: influenzae, make the last letter (e) italic as well.

13. Line 210: form or from?

With regards,

6. PLOS authors have the option to publish the peer review history of their article (what does this mean?). If published, this will include your full peer review and any attached files.

Reviewer #1: **Yes: **Raad N Hasan

Reviewer #2: No

Reviewer #3: No

---

## [Author Response · Author response to Decision Letter 0]

9 Sep 2022

PONE-D-22-18329

Disinfection of otorhinolaryngological endoscopes with electrolyzed acid water: a cross-sectional and multicenter study

PLOS ONE

Dear Dr. Awatif Abid Al-Judaibi,

Academic Editor

PLOS ONE

We would like to thank you and the Reviewers for the thought-provoking comments on our manuscript. We have revised the paper in accordance with the comments made by the Reviewers, and made our point-by-point responses as follows.

Journal Requirements:

＞ Thank you so much for your comment. We have read through two of the PLOS ONE style templates and confirm that our manuscript meets PLOS ONE's style requirements.

"I have read the journal's policy and the authors of this manuscript have the following competing interests: Protein Clean Sheets and a CLEANTOP KD-1 automatic washer and disinfector used in the present study were provided by Kaigen Pharma, Co., Ltd. and one of the authors, Hideki Hayashi, is an employee of Kaigen Pharma, Co., Ltd."

＞ We have included the statement "This does not alter our adherence to PLOS ONE policies on sharing data and materials.” in the cover letter.

> Thank you for your suggestion. We have reviewed the reference list again and confirm that it is correct.

Additional Editor Comments:

Reviewer 1:

1- Significant differences between A-1, A-2, and A-3. sampling sites should be mentioned.

> We really appreciate the reviewer's suggestion. As this comment would be related to reviewer 3's comment No. 4, we have edited the new Fig. 1 to clearly demonstrate the sampling sites.

2- There are some typos and grammatical errors which should be revised.

> Thank you for your careful reading. We have revised the manuscript again, and also used an English language editing service provided by Editage (www.editage.com).

3- This sentence is not correct "The verbal informed consent for participation was obtained form patients visiting the outpatient facility of the

Department of Otorhinolaryngology using an opt-out methodology because this study does not harm the patients in any way including direct microbiological sampling or blood sampling"

> This comment would be related to reviewer 3's comment No. 13. We fixed the typo "form" into "from". We also rewrote the sentence for clarity (Page 13, line 221 - 224).

4- I couldn't find any of the references from the 21, 22 and 23

Reference correction is necessary

> We appreciate careful review by Reviewer 1. We believe this is not the case. Reference 21 was cited in line 157, 22 was cited in line 189, 23 was cited in line 196, in the former form of manuscript. In the current form of manuscript, reference 22 (changed from previous reference 21) is cited in Page 10, line 169, reference 23 (changed from previous reference 22) is cited in Page 12, line 201, and reference 24 (changed from previous reference 23) is cited in Page 12, line 209.

Reviewer 2:

Authors have reported effective use of Electrolyte acid water (EAW) in disinfecting otorhinolaryngological endoscopes, which is of clinical significance. Authors have presented sufficient data in support of research question keeping limitation of study in account. Manuscript can be accepted for publication after making amendments as suggested.

> We really appreciate the reviewer's careful and fruitful comments.

Some specific comments/reviews to incorporate/amend are as follows:

1- Abstract: Authors have summarized key findings appropriately. However, in vitro inactivation of bacteria and SARS-CoV-2 need to be presented consistently in number/percentage or CFU to enhance reader's understanding.

> Thank you for the important comment. We rewrote the abstract and added data in number, ratio, and CFU (Page 3, line 41 - 43).

2-Introduction: Significance of using EAW for disinfection of otorhinolaryngological endoscopes for bacteria has been presented appropriately. however, authors need to include significance of using EAW for SARS-CoV-2 as well briefly.

> We really appreciate the reviewer's suggestion. We added the part describing relationship between endoscope reprocessing and SARS-CoV-2 (Page 5, line 73 - 77).

3- Methods: Authors used appropriate methodology to address the research question. However, following amendments 13are suggested:

Page 8, lines 138-139: Authors either need to include time/concentration for disinfection using Cleantop-KD-1 or reference for the procedure.

> We thank the reviewer for raising this point. We added time and concentration for disinfection with Cleantop-KD-1 (Page 9, line 150 -152).

Page 11, lines 190-191: "Overnight" incubation time need to be rephrased in terms of "Hours" for clarity.

> We also rephrased "for 16 h" instead of "overnight" for clarity (Page 12, line 197).

4- Results: Authors have presented results appropriately using tables facilitating reader's understanding. Suggested amendment:

Page 13, lines 220-221: "29 patients had nasal disease....." needs correction as breakdown adds up to 30.

> We really appreciate the reviewer's indication. As related to the Reviewer 3's comment No. 7, we rewrote the number of patients with sinusitis, and added the new table 1 for clarity (Page 14, line 240). According to the addition of new table 1, previous table 1, 2, 3, and 4 were renumbered into the new table 2, 3, 4, and 5, respectively.

Page 14, lines 234- 235: " ...only in 1 of 75 patients...." needs to be corrected as " ...only in 1 of 75 endoscopes...."

> According to the reviewer's suggestion, we changed "patients" into "endoscope" (Page 15, line 249).

5-Discussion: Overall needs to be rephrased. Few amendments suggested are:

Page 18-19, line 286- 301, "Because otorhinolaryngology ........high-level disinfection." is a repeat from introduction need to be deleted.

> Thank you so much for suggestion. This comment would be related to the reviewer 3's comment No. 10. We removed the indicated part from the section of Discussion. 

Page 19-20, para 2, line 309-320: Needs rephrasing for clarity. Also, reference of table needs to be removed from discussion.

> We thank the reviewer for suggestion. We rewrote this part in the section of Discussion for clarity (Page 20, line 301 - Page 21, line 318). In addition, we removed the reference of table 3 in this part.

6- Conclusion: It is vague and needs rephrasing keeping context of study in view to highlight the significance of study.

> We appreciate the reviewer's suggestion. This comment would be related to the reviewer 3's comment No. 11. We totally rephrased the section of Conclusion for clarity according to the reviewers' suggestion (Page 23, line 349 - 358).

Reviewer 3:

1. Lines 90 - 93: Despite there is no study that presents the disinfection efficacy of EAW in clinically used otorhinolaryngological endoscopes, authors have mentioned about the effectiveness of using EAW in disinfecting clinically for gastrointestinal endoscopes. Authors should highlight their significant findings and contributions.

> Thank you so much for your constructive suggestion. We added a few sentences to emphasize the disinfection efficacy of EAW for gastrointestinal endoscopes highlighting findings shown by the previous study (Page 6, line 96 - 102).

2. Introduction can be improved by adding a few sentences about the hypochlorous acid since it is the primary antibacterial agent in the study.

> This comment might be related to the next comment. We totally agree with the comment and added the phrases to the part regarding hypochlorous acid (Page 5, line 86 - Page 6, line 91).

3. Line 95: "Fig 1. Schematic of generation of electrolysed acid water" is not necessary to be included in the manuscript, as it is well known.

> According to the reviewer's suggestion, we removed "previous Fig 1" that shows generation of EAW. Instead, we added the new Fig 1 to show the sampling sites in endoscopes (Page 8, line 141).

4. Lines 116 –148: it is helpful to provide an image (as supporting Information file) of the endoscopy instruments with two sites (A-1 to A-3, B-1 to B-3) to aid the understanding of this part.

> We really appreciate the reviewer's suggestion. This comment would be related to reviewer 1's comment No. 1. We accordingly have edited the new Fig. 1 to clearly demonstrate the sampling sites (Page 8, line 141).

5. Line 149 and line 340: this a study designed to culture aerobic and anaerobic bacteria as well as fungi but there is limited discussion about fungi being observed or detected or fungicidal effect, please explain more.

> This suggestion was very helpful for us. We added a few phrases regarding fungi in the section of Introduction (Page 6, line 92) and Results (Page 17, line 269) using reference 16 and 17. We renumbered the references cited in the later part of the new manuscript, accordingly. In addition, we added a few sentences to explain the fungicidal effect of EAW in the section of Discussion (Page 21, line 323 - 326).

6. Line 171: the subheading "bacteria and culture" is more generic while the paragraph discusses EAW treatment; please edit the subheading to reflect all the aspects of the method on bacterial culture and bactericidal effects of EAW.

> According to the reviewer's suggestion, we changed the subheading "bacteria and culture" into "bacterial culture and bactericidal effects of EAW" (Page 11, line 184).

7. Lines 216- 226: A table is preferred here given the amount of data being presented.

> We really appreciate the reviewer's indication. As this comment would be related to the Reviewer 2's comment No. 4, we rewrote the number of patients with sinusitis, and added table 1 for clarity (Page 14, line 240).

8. Line 270: What is the control sample for the data presented in the Table 3?

> Actually, we didn't set the control sample for data in new Table 4 (previous Table 3). Instead, we added the sentence to this part as follows; " No controls were used in this study, however, we confirmed that microorganisms not treated with EAW survived at more than 107 under the same culture conditions." (Page 18, line 281 - 283).

9. Full name of bacterial strains should be mentioned in the (Table 3) as mentioned in from line 172-175 have to be.

> According to the reviewer's comment, we added full name of bacterial strains in the new Table 4 (previous Table 3) as mentioned in from line 185 -188 in the current form of manuscript.

10. Lines 258 – 301: Discussion can be improved to avoid repetition with introduction section.

> Thank you so much for suggestion. This comment would be related to the reviewer 2's comment No. 5, we removed the indicated part from the section of Discussion. 

11. The conclusion is supported by the results for EAW treatment of SARS-CoV-2 but not for bacteria. Therefore, this section must be improved and highlight why EAW can be an ideal disinfectant (bactericidal and virucidal effects) and use it at present as regarded to be safe and environmentally friendly to overcome supply crisis during pandemic emergency and enable the safe continuity.

> We really appreciate the reviewer's suggestion. This comment would be related to the reviewer 2's comment No. 6, we totally rephrased the section of Conclusion for clarity according to the reviewers' suggestion (Page 23, line 349 - 358).

12. Line 183: influenzae, make the last letter (e) italic as well.

> Thank you for your indication. We made the last latter (e) italic accordingly.

13. Line 210: form or from

> This comment would be related to reviewer 1's comment No. 3. We fixed the typo "form" into "from".

---

## [Editor Report · Decision Letter 1]

19 Sep 2022

Disinfection of otorhinolaryngological endoscopes with electrolyzed acid water: a cross-sectional and multicenter study

PONE-D-22-18329R1

Dear Dr. Koichi Omori,

We’re pleased to inform you that your manuscript has been judged scientifically suitable for publication and will be formally accepted for publication once it meets all outstanding technical requirements.

Kind regards,

Awatif Abid Al-Judaibi, PhD

Academic Editor

PLOS ONE

---

## [Editor Report · Acceptance letter]

23 Sep 2022

PONE-D-22-18329R1 

Disinfection of otorhinolaryngological endoscopes with electrolyzed acid water: a cross-sectional and multicenter study 

Dear Dr. Omori:

I'm pleased to inform you that your manuscript has been deemed suitable for publication in PLOS ONE. Congratulations! Your manuscript is now with our production department. 

Kind regards, 

on behalf of

Professor Awatif Abid Al-Judaibi 

Academic Editor

PLOS ONE